# General practitioners providing non-urgent care in emergency department: a natural experiment

Olalekan A Uthman,[1] Clare Walker,[1] Sudakshina Lahiri,[2] David Jenkinson,[1] Victor Adekanmbi,[1,3] Wendy Robertson,[1] Aileen Clarke[1]

## ABSTRACT

**Objective** To examine whether care provided by general practitioners (GPs) to non-urgent patients in the emergency department differs significantly from care provided by usual accident and emergency (A&E) staff in terms of process outcomes and A&E clinical quality indicators.

**Design** Propensity score matched cohort study.

**Setting** GPs in A&E colocated within the University Hospitals Coventry and Warwickshire NHS Trust between May 2015 and March 2016.

**Participants** Non-urgent attendances visits to the A&E department.

**Main outcomes** Process outcomes (any investigation, any blood investigation, any radiological investigation, any intervention, admission and referrals) and A&E clinical indicators (spent 4 hours plus, left without being seen and 7-day reattendance).

**Results** A total of 5426 patients seen by GPs in A&E were matched with 10 852 patients seen by emergency physicians (ratio 1:2). Compared with standard care in A&E, GPs in A&E significantly: admitted fewer patients (risk ratio (RR) 0.28, 95% CI 0.25 to 0.31), referred fewer patients to other specialists (RR 0.31, 95% CI 0.24 to 0.40), ordered fewer radiological investigations (RR 0.38, 95% CI 0.34 to 0.42), ordered fewer blood tests (0.57, 95% CI 0.52 to 0.61) and ordered fewer investigations (0.93, 95% CI 0.90 to 0.96). However, they intervened more, offered more primary care follow-up (RR 1.78, 95% CI 1.67 to 1.89) and referred more patients to outpatient and other A&E clinics (RR 2.29, 95% CI 2.10 to 2.49). Patients seen by GPs in A&E were on average less likely to spend 4 hours plus in A&E (RR 0.37, 95% CI 0.30 to 0.45) compared with standard care in A&E. There was no difference in reattendance after 7 days (RR 0.96, 95% CI 0.84 to 1.09).

**Conclusion** GPs in A&E tended to manage self-reporting minor cases with fewer resources than standard care in A&E, without increasing reattendance rates.

## Strengths and limitations of this study

► Our results have important clinical implications as they inform optimal care for non-urgent care patients in accident and emergency.
► The main limitation of the study is the lack of random allocation and as such our findings may be prone to selection bias.
► We do not know the eventual health outcome, although reattendance data are reassuring on this front.

## BACKGROUND

Accident and emergency (A&E) departments in the UK offer access 24 hours a day, 365 days a year, and the immediate team usually includes paramedics, A&E nurses, diagnostic radiographers, A&E reception staff, porters, healthcare assistants and emergency medicine doctors.[1] Following arrival by ambulance or other self-determined mode of transport, the patient is triaged by a qualified healthcare professional to ensure that patients with the most serious conditions are seen first.[1] Depending on the presenting symptoms and signs, patients can either be treated, transferred to a nearby urgent care centre, minor injuries unit, referred to a general practitioner (GP) on site or discharged.[1] Where treated, a patient is cared for by a team headed by a consultant in emergency medicine and the patients receive care from a suitably qualified member of the team.

A&E departments in the UK are facing a crisis in terms of overcrowding with long waiting times. Numerous factors have been suggested for this overcrowding in the use of A&E services, including 'inappropriate' use by patients with non-urgent conditions. A wide range (between 6.7% and 89%) of A&E visits has been classified as non-urgent depending on definition; and it is thought that many patients could have their care needs met in non-hospital (including in GP) settings.[2–6] Inappropriate use is thought to compromise A&E services for true emergencies.[7 8] And it has been suggested that non-urgent cases might be managed hastily or may not be linked in with the comprehensive and continuous care available in the primary care setting, thus leading to suboptimal care.[2]

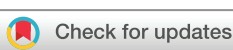

[1]Division of Health Sciences, Warwick Medical School, The University of Warwick, Coventry, UK
[2]Institute of Digital Healthcare, WMG, The University of Warwick, Coventry, UK
[3]Division of Population Medicine, Cardiff University, Cardiff, UK

**Correspondence to**
Prof Aileen Clarke;
aileen.clarke@warwick.ac.uk

Reducing inappropriate use of A&E has been an important area for policy makers for a long time,[9–12] and experts have recently recommended that 'All hospital emergency departments in the United Kingdom should include a primary care out-of-hours facility to help ease the ongoing pressure on the system'.[13] The 'introduction of GPs may provide more comprehensive and cost- and resource-effective care for patients with non-urgent problems in the ED' it is thought.[14] It has also been suggested that GPs in A&E could reduce waiting times by seeing non-urgent patients promptly and freeing regular A&E physicians to see more urgent cases.[15]

Previous studies have compared the effects of introducing GPs in A&E with standard care in A&E for managing non-urgent cases.[14 16–18] However, these studies have focused on process outcomes only. To our knowledge, no studies have examined the impact of GP in A&E on A&E clinical indicators. The present evaluation deals with a critical research gap highlighted by a previous Cochrane Review,[14] described as the need for further studies of the effects of GP in A&E 'on wait times, adverse effects, mortality and patient outcomes'. The main purpose of this study was therefore to examine whether care provided to non-urgent patients by GPs in the A&E differed significantly from care by usual A&E staff in terms of process outcomes and A&E clinical quality indicators.

## METHODS
### Study design, setting and source of data
We used anonymised individual-level acute care activity data from the University Hospitals Coventry and Warwickshire NHS Trust (UHCW) between May 2015 and March 2016. We used the Urgent Care Project Board definition[19] to identify non-urgent attendances (self-reporting minor cases). These are unplanned visits to the A&E defined as those presenting with minor illness with a National Early Warning Score ≤1. The three relevant Healthcare Resource Group (HRG) codes are: VB08Z (Emergency Medicine, category 2 investigation with category 1 treatment), VB09Z (Emergency Medicine, category 1 investigation with categories 1–2 treatment) and VB11Z (Emergency Medicine, no investigation with no significant treatment).

### Intervention evaluated: GP in A&E scheme
The intervention of interest was whether the patients received care under the Secondary Care A&E team or by a GP in A&E. The GP in A&E scheme was set up by the Coventry and Rugby GP Alliance as part of the Prime Minister's GP Access Fund (PMAF) initiative. PMAF was a £50m Challenge Fund to support practices to try out new and innovative ways of delivering GP services and making services more accessible to patients.

The patients arriving at the A&E were registered onto the Trust's clinical A&E system and triage was undertaken initially by the A&E triage nurse and by either an advanced nurse practitioner or a senior nurse, who also undertook an initial visual assessment of the patient. Patients suitable for GP in A&E were identified and those deemed unsuitable were returned to existing A&E clinical pathways. Staff in the GP in A&E service also actively identified cases based on observation of patients in A&E, and/or based on information recorded in patient registration notes. Patients were not randomised. Once the decision to transfer to GP in A&E was made, the patient was moved onto the Trust's clinical A&E system by a receptionist and into the GP in A&E clinic. Patients suitable for the service were then diagnosed and treated by a GP in A&E. The patient's GP practice notes (accessed via the primary care shared record) were available during the consultation. Patients who needed further tests or admission to hospital were referred to UHCW existing hospital services. Onward referral postdischarge to a registered GP was via access to the primary care shared record. The GP in A&E scheme operated 7 days a week, for 12 hours per day at peak times.

### Outcomes and covariates
The outcomes of interest were:
1. Process outcomes: any investigations, any blood investigations, any radiological investigations, any intervention (including prescriptions) and admissions, discharges, referrals to outpatient clinics or other A&E clinics, referrals to other specialists and follow-up treatment to be provided by the GP in primary care setting.
2. A&E clinical indicators: percentage of patients who spent 4 hours plus, who left without being seen and 7-day reattendance rates.

Confounding variables included in the analyses were patients' age, sex, ethnicity, number of procedures, distance to UHCW (partial post code), number of presenting symptoms and arrival hour.

### Statistical analyses
We examined the baseline characteristics of the patient cohort and estimated standardised differences for all variables before and after matching. A standardised difference of 10% or more is suggestive of imbalance.[20] We used propensity score[21] methods to account for all measured differences in baseline characteristics between patients seen by the GP in A&E and standard care in A&E. The propensity score approach was used to control for all observed confounding factors that might influence assignment and outcome. We constructed a sample of patients balanced on covariates and risk factors (such as age, sex, ethnicity, number of procedures and number of presentations). We constructed the propensity scores using a logistic regression. We then matched each patient seen by GP in A&E to standard care in A&E patients with the closest propensity score on a ratio of 1:2 using a nearest neighbour algorithm with no replacement. We calculated the average treatment effect on treated patients, which is a measure of the impact of GP in A&E on process outcomes and A&E clinical indicators. To test

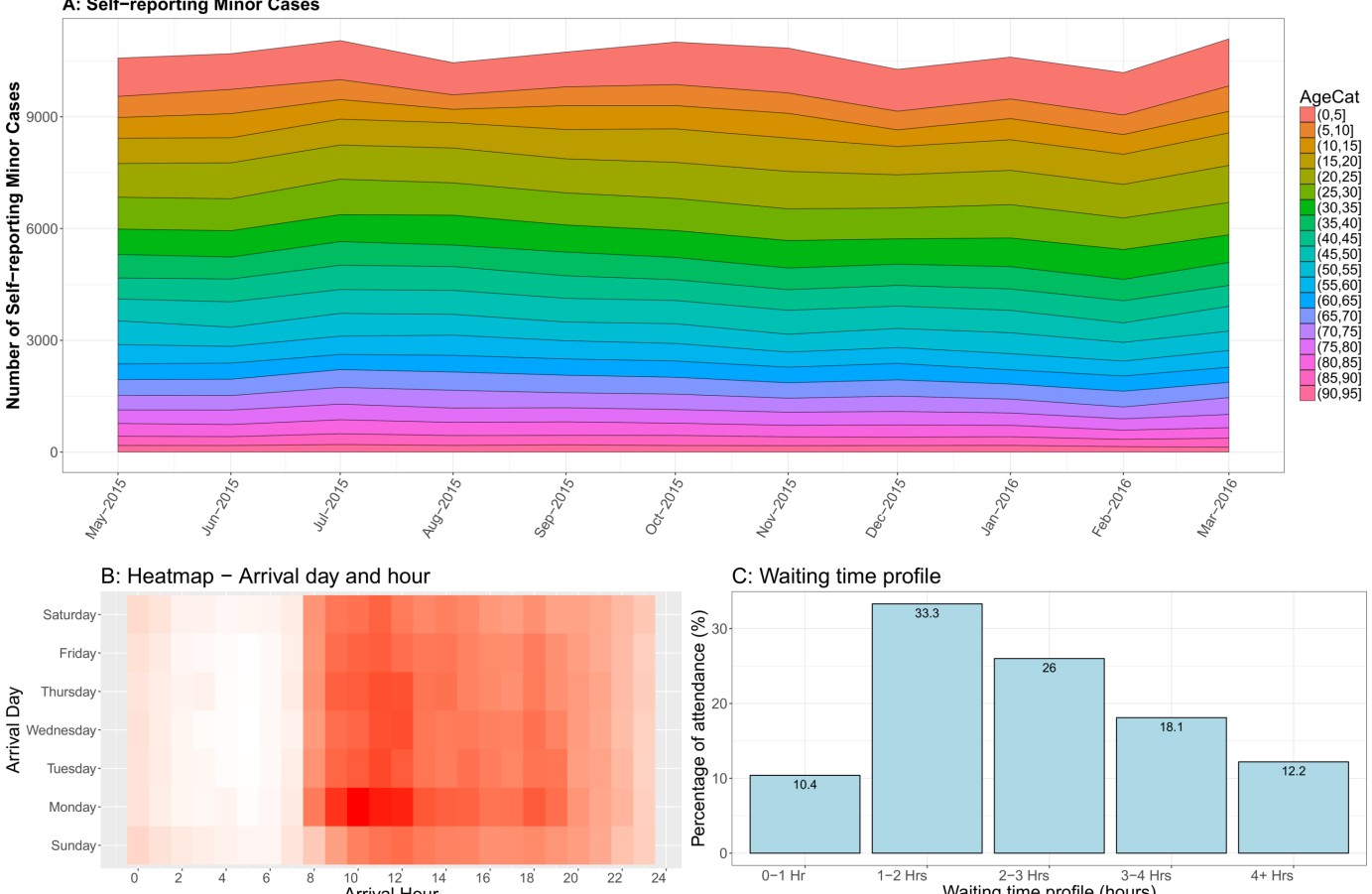

**Figure 1** Summary characteristics.

the robustness of our primary findings, we performed a series of sensitivity analyses. First, we repeated the base case algorithm, nearest neighbour with replacement. Second, we also implemented another algorithm 'exact' match with and without replacement. All data were analysed using R statistical software V.3.4.0. The null hypothesis was tested against a two-sided alternative hypothesis at a significance level of 5%.

## Patient involvement

The Collaborations for Leadership in Applied Health Research and Care (CLAHRC) West Midlands Patient and Public Involvement (PPI) advisors reviewed the study on a number of occasions. They were involved with the design of the research question and gave feedback at our advisory committee meeting. The research question was generated by the Coventry GP Alliance board who were the funders of the study. The results were discussed with PPI advisors and local authorities at a large stakeholder event. The results have been disseminated via our local authorities and shared through patient and public involvement initiatives.

## RESULTS
### Patients and characteristics
A total of 170 154 patients attended UHCW A&E between May 2015 and March 2016 (figure 1A). Our primary

cohort consisted of 120 034 self-reporting minor case attendances in A&E at UHCW. Of every 10 attendees, seven were self-reporting minor cases during this period. Attendances were relatively even throughout the week, with the highest attendances on a Monday (approximately 16%). Attendance was greatest on each day between 09:00 and 11:00 (figure 1B). Most patients (70%) were dealt with within 3 hours of attendance at A&E (figure 1C).

Characteristics of the patients before and after matching are summarised in table 1 (and online supplementary eTables 1, 2 and 3). There were important baseline differences between patients seen by GPs in A&E and those seen in the normal pathway. GPs in A&E tended to see fewer white patients (69.9% vs 80.1%). They also tended to see patients who lived closer to the hospital; 43.3% of GP in A&E patients lived within a 5-mile radius of the A&E compared with 30.1% in standard care in the A&E group. We successfully matched 5426 patients seen by GPs in A&E with 10 852 standard care in A&E patients. After matching, absolute standardised differences were less than 10% for all variables used for the propensity score matching, suggesting an adequate match.

### Process outcomes
Process outcomes and A&E clinical indicators are presented in figure 2. GPs in A&E admitted significantly

**Table 1** Basic characteristics of the patients included in the analyses

| | Before matching | | | | After matching | | | |
|---|---|---|---|---|---|---|---|---|
| | Regular emergency physicians (n=114 608) | GP in A&E (n=5426) | P values | Standardised difference | Regular emergency physicians (n=10852) | GP in A&E (n=5426) | P values | Standardised difference |
| Age mean, SD | 36.6 (26.2) | 37.2 (18.9) | 0.126 | 0.550 | 37.03 (19.2) | 37.2 (18.9) | 0.755 | 0.099 |
| Male (%) | 56 860 (49.6) | 2488 (45.9) | | −0.038 | 5001 (46.1) | 2488 (45.9) | 0.794 | −0.002 |
| Ethnicity (%) | | | <0.001 | | | | 0.271 | |
| White | 91 760 (80.1) | 3794 (69.9) | | −0.101 | 7684 (70.8) | 3794 (69.9) | | −0.009 |
| Asian | 10 904 (9.5) | 699 (12.9) | | 0.034 | 1401 (12.9) | 699 (12.9) | | 0.000 |
| Black | 4023 (3.5) | 294 (5.4) | | 0.019 | 545 (5.0) | 294 (5.4) | | 0.004 |
| Mixed | 2603 (2.3) | 117 (2.2) | | −0.011 | 187 (1.7) | 117 (2.2) | | 0.004 |
| Others | 5318 (4.6) | 522 (9.6) | | 0.050 | 1035 (9.5) | 522 (9.6) | | 0.001 |
| Distance (%) | | | <0.001 | | | | 0.844 | |
| <5 miles | 34 491 (30.1) | 2351 (43.3) | | 0.009 | 4711 (43.4) | 2351 (43.3) | | 0.000 |
| 5–10 miles | 48 261 (42.1) | 2237 (41.1) | | −0.008 | 4492 (41.4) | 2237 (41.1) | | −0.002 |
| 10 to 20 miles | 24 721 (21.6) | 504 (9.3) | | −0.123 | 1014 (9.3) | 504 (9.3) | | −0.001 |
| 20 to 50 miles | 4763 (4.2) | 156 (2.8) | | −0.013 | 272 (2.5) | 156 (2.8) | | 0.003 |
| >50 miles | 2372 (2.1) | 181 (3.3) | | 0.013 | 363 (3.3) | 181 (3.3) | | 0.000 |
| Presenting symptoms (%) | | | <0.001 | −0.099 | | | 0.131 | 0.008 |
| 0 | 57 317 (50.0) | 3196 (58.9) | | | 6408 (59.0) | 3196 (58.9) | | |
| 1 | 52 090 (45.5) | 2030 (37.4) | | | 4094 (37.7) | 2030 (37.4) | | |
| 2 | 4686 (4.1) | 185 (3.4) | | | 336 (3.1) | 185 (3.4) | | |
| 3 or more | 515 (0.4) | 15 (0.3) | | | 14 (0.1) | 15 (0.3) | | |
| Arrival hour mean, SD | 13.52 (5.51) | 14.07 (3.52) | <0.001 | 0.5501 | 14.07 (4.62) | 14.07 (3.52) | 0.998 | −0.000 |

A&E, Accident and Emergency; GP, general practitioner.

fewer patients with self-reporting minor cases (301/5426, 5.5%) than standard care in A&E (2168/10852, 20.1%; RR 0.28, 95% CI 0.25 to 0.31).

GPs in A&E referred fewer patients to other specialists (72/5426, 1.3%) than standard care in A&E (465/10852, 4.3%, risk ratio (RR) 0.31, 95% CI 0.24 to 0.40). GPs in

| Outcome | % GP–in–ED (n=5,426) | % Regular ED Care (n=10,852) | Risk Ratio | RR | 95%–CI |
|---|---|---|---|---|---|
| **Process outcome** | | | | | |
| Any admission | 5.5 | 20.1 | | 0.28 | [0.25; 0.31] |
| Referrals to other specialists | 1.3 | 4.3 | | 0.31 | [0.24; 0.40] |
| Any radiological investigation | 7.2 | 18.9 | | 0.38 | [0.34; 0.42] |
| Any blood investigation | 12.1 | 21.5 | | 0.57 | [0.52; 0.61] |
| Any investigation | 45.9 | 49.3 | | 0.93 | [0.90; 0.96] |
| Any intervention (including prescription) | 35.6 | 27.5 | | 1.29 | [1.23; 1.35] |
| Follow–up with primary care physician | 28.5 | 16.0 | | 1.78 | [1.67; 1.89] |
| Referrals to out–patient / other A&E clinic | 18.5 | 8.1 | | 2.29 | [2.10; 2.49] |
| **A & E clinical indicator** | | | | | |
| Spent for 4–hour plus | 1.9 | 5.3 | | 0.37 | [0.30; 0.45] |
| Left without being seen | 2.2 | 3.9 | | 0.57 | [0.46; 0.69] |
| Seven day re–attendance | 5.5 | 5.7 | | 0.96 | [0.84; 1.09] |

Favours GP–in–ED    Favours Regular ED care

**Figure 2** Process outcomes and A&E clinical indicators. A&E, accident and emergency; ED, emergency department; GP, general practitioner; RR, risk ratio.

A&E were significantly less likely than standard care in A&E to order radiological investigations (RR 0.38, 95% CI 0.34 to 0.42) and blood tests (RR 0.57, 95% CI 0.52 to 0.61) for patients. Also, on average, GPs in A&E ordered significantly but slightly fewer investigations overall, than standard care in A&E (RR 0.93, 95% CI 0.90 to 0.96). GPs in A&E intervened more (RR 1.29, 95% CI 1.23 to 1.35) and referred significantly more patients to outpatients and other A&E clinics (RR 2.29, 95% CI 2.10 to 2.49) than standard care in A&E. GPs in A&E also discharged more patients with the offer of significantly more primary care follow-up than standard care in A&E (RR 1.78, 95% CI 1.67 to 1.89).

### A&E clinical indicators

The proportion of patients (all self-reporting minor cases) who spent 4 hours plus in the A&E was statistically significantly lower for patients seen by the GPs in A&E (105/5426, 1.9%) than for those seen by standard care in A&E (573/10852, 5.3%; RR 0.37, 95% CI 0.30 to 0.45). Patients assigned to GP in A&E were less likely to leave without being seen (121/5426, 2.2%) compared with those assigned to standard care in A&E (427/10852, 3.9%; RR 0.57, 95% CI 0.46 to 0.69). The rate of 7-day reattendance was similar among those patients seen by GP in A&E (296/5426, 5.5%) and emergency physicians (619/10852, 5.7%; RR 0.96, 95% CI 0.84 to 1.09).

### Sensitivity analyses

The results of the sensitivity analyses for process outcomes and A&E clinical indicators are shown in online supplementary eFigures 1 and 2, respectively. The magnitudes and directions of the effect estimates were very similar to the primary analyses for both sensitivity analyses: the replacement method and an alternative matching algorithm 'exact' with and without replacement. For example, the RR for any admission ranged from 0.27 to 0.29: nearest neighbour without replacement (RR 0.28, 95% CI 0.25 to 0.31), nearest neighbour replacement (RR 0.29, 95% CI 0.26 to 0.32), 'exact' algorithm without replacement (RR 0.27, 95% CI 0.24 to 0.30) and 'exact' algorithm replacement (RR 0.27, 95% CI 0.24 to 0.30).

### DISCUSSION
### Main findings

We found significant differences between GPs in A&E and standard care in A&E. GPs in A&E referred fewer patients to other specialists, admitted fewer patients, ordered fewer radiological investigations, fewer blood tests and fewer investigations overall and intervened more often. They offered more primary care follow-up and referred more patients to outpatient and other A&E clinics. More importantly, on average, patients seen by GPs in A&E: were less likely to spend 4 hours plus in A&E and were no more likely to reattend after 7 days. The findings corroborate those of the previous three non-randomised studies[16–18] included in a Cochrane review.[14]

---

> **Box 1  Models of primary care services in accident and emergency (A&E)[23]**
>
> 1. Redirect—Patients present to the A&E and are sent to a primary care service:
>    a. Adjacent out-of-hours service.
>    b. Adjacent walk-in-centre.
>    c. Adjacent primary care/community service.
>    d. Advice only/self-care.
> 2. Managing patients in the emergency department:
>    a. Gatekeeping in A&E—Primary care service based at the front of A&E to manage patient entry to the A&E service.
>    b. Primary care within A&E:
>       i. General practitioner working in A&E:
>          1. Employed by primary care trust.
>          2. Employed by Acute Trust.
>       ii. Other primary care clinician.
>       iii. A&E clinician.

The review involved more than 10 000 non-urgent cases from 52 A&E units, and provided evidence that GPs in A&E ordered fewer X-rays and admitted fewer patients compared with standard care in A&E. Our study also showed that GPs prescribed oral medication and to take out (TTO) medication more often, more oral antibiotics, more topical creams and more ear drops. More TTOs may mean they are writing more repeat prescriptions or change existing prescriptions to optimise therapy. GPs prescribe less eye drops in general, slightly less rectal medication and non-steroidal anti-inflammatory drugs, although they gave verbal guidance at similar rates as standard care in A&E.

As most of the A&E workload takes place from Monday to Friday and between 08:00 and 22:00, there is a case for extending coverage of GP in A&E beyond these hours but not to 24 hours a day. Demand for primary care consultations at A&E locations reduce during the night and alternative primary care services are available. NHS England's chief executive and the chief executive of NHS Improvement recently recommended that 'Every hospital in England must have a 'comprehensive' GP led triage system in emergency departments to avoid a repeat of the winter crisis that gripped the service this year'.[22] Two broad models of primary care services in A&E have been proposed[23] and these models are summarised in box 1[23]:

Our findings relate most closely to model 1—a redirect model with an adjacent primary care service. Another recent systematic review examined the effectiveness of different models of delivering urgent care.[24] The review included 45 systematic reviews and 102 primary studies and found that the evidence on colocation of GP services with A&Es indicates that there is potential to improve care.[24]

There are several workforce implications, however. Many organisations are seeking to train and recruit GPs who can work in acute medicine settings in secondary and primary care centres with the expectation of developing a cadre of multiskilled healthcare professionals.[25] This is in

keeping with model 2 with fully integrated primary care professionals in the A&E department. There is perhaps also a case for extending opportunities for secondary care A&E physicians to work in primary care during their training to develop some of the GP skillset which has less reliance on tests and a greater experience and knowledge of which cases can safely be managed outside of the hospital.[26] However, recruitment and retention pressures in both specialities make this kind of collaborative training unlikely at present.

There is also the opportunity for the primary care approach to diffuse into the A&E—this is less likely in a self-contained, although colocated, unit than locations where primary and secondary care clinicians work in a more integrated fashion.

There may be a dilution of primary care skills and approaches over time for a GP fully integrated in A&E and their effectiveness will decrease over time as they become subsumed into the secondary care culture. However, if secondary care seek to recruit less experienced GPs to fill these roles, this may be more of a problem, as they may not have sufficient primary care experience as an independent practitioner to maintain their professional skills and identity in a secondary care environment. A GP in A&E requires an accommodation of a culture which is used to handling uncertainty with a culture which seeks to reduce uncertainty in diagnosis and treatment of patients.

### Study limitations and strengths

The main limitation of the study is the lack of random allocation.[27 28] Our findings may be prone to selection bias. Although we used a propensity score matching method to control for known baseline differences, it is possible that we have not controlled for important confounders, such as severity of presentation. We did however adjust our analysis for number of presenting symptoms, number of procedures conducted and the number of tests ordered. In addition, we do not know the eventual health outcome, although reattendance data are reassuring on this front.

Notwithstanding these limitations, the findings have some important policy and practice implications. Consistent with previous studies,[16–18 29] these findings provide empirical evidence about GPs in A&E and A&E clinical indicators, supporting theories that 'the introduction of GPs may provide more comprehensive and cost- and resource-effective care for patients with non-urgent problems in the A&E'.[14] The 2016 and 2017 'extraordinary pressures' on A&E services across the country and schemes like this one will play a part in relieving this pressure. Our findings suggest that GPs are offering more comprehensive care. We hypothesise that the fact that the GPs in A&E had access to primary care notes may be of importance here. Findings show that GPs are more likely to make immediate onward referral to ambulatory clinics such as the urgent primary care clinic in UHCW or secondary care outpatient clinics. This is a more efficient use of resources avoiding both hospital admission and referral back to usual GP for them to make the referral.

The removal of this unnecessary additional interaction reduces frustration for patients and doctors alike and should improve the patient experience. In addition, the prescribing data we examined could mean that GPs in A&E are fulfilling requests for repeat medications more often than A&E staff with the added safety of access to the primary care record again reducing the need for additional communications with GPs.

We used data from one of the largest acute teaching hospitals in the UK. Our propensity score approach allowed us to reduce (though not eliminate) the possibility of confounding in relation to several relevant aspects of severity of presentation, as well as allowing us to balance a wide range of sociodemographic characteristics. The approach is consistent with Medical Research Council guidelines for using natural experiments to evaluate population health interventions.[30]

### CONCLUSION

In this observational study of patients self-reporting with minor illness to the A&E department in one large teaching hospital, GPs in A&E made fewer referrals to other specialists, admitted fewer patients, ordered fewer radiological investigations, fewer blood tests and fewer investigations overall and intervened more often, compared with standard care in A&E. Overall, they managed patients with fewer resources than standard care in A&E, without increasing reattendance rates.

**Acknowledgements** Work as part of the evaluation of the Best Care, Anywhere: Integrating Primary Care in Coventry Programme, which was essentially a collaborative effort between the Coventry–Rugby GP Alliance, the University Hospital Coventry and Warwickshire and academic partners, the University of Warwick, Coventry University and University of Birmingham. We would like to take this opportunity to express our gratitude to all members of the Coventry–Rugby GP Alliance Steering Group for their thoughtful discussions with the research team regarding the multifaceted nature of the challenges currently facing the primary and acute care sectors in the region. We also gratefully acknowledge the contributions of University Hospital Coventry and Warwickshire for their efforts in sharing hospital activity data with the research team which made this study possible. Dr Wendy Robertson, Dr Victor Adekanmbi, Dr David Jenkinson and Professor Aileen Clarke are partly funded by the NIHR CLAHRC West Midlands initiative. Dr Clare Walker holds an NIHR funded ACF post. Dr Olalekan A. Uthman is supported by the National Institute of Health Research using Official Development Assistance (ODA) funding. This report presents independent research and the views expressed are those of the authors and not necessarily those of the NHS, the NIHR or the Department of Health and Social Care.

**Contributors** OAU, CW, SL, DJ, WR and AC were responsible for study concept and design. OAU and SL were responsible for acquisition of data. OAU, CW and AC drafted the manuscript. OAU, SL and VA performed the statistical analysis. All authors interpreted the data, critically revised the manuscript for important intellectual content and approved the final version for submission. All authors agree to be accountable for all aspects of the work. OAU and AC are guarantors.

**Funding** Coventry–Rugby GP Alliance.

**Competing interests** None declared.

**Patient consent** Not required.

**Ethics approval** The University of Warwick Biomedical Science Research Ethics Committee approved the study.

**Provenance and peer review** Not commissioned; externally peer reviewed.

**Data sharing statement** No additional data available.

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
