## [Reviewer comments · BMJ Open]

ARTICLE DETAILS

TITLE (PROVISIONAL)	General practitioners providing non-urgent care in emergency department: a natural experiment
AUTHORS	Uthman, Olalekan; Walker, Clare; Lahiri, Sudakshina; Jenkinson, David; Adekanmbi, Victor; Robertson, Wendy; Clarke, Aileen

VERSION 1 – REVIEW

REVIEWER	Colin O'Keeffe School of Health and Related Research University of Sheffield United Kingdom
REVIEW RETURNED	27-Oct-2017

GENERAL COMMENTS	This was a well written, clearly described paper on a key area/topic for the NHS. The conclusions are valid and adequately discuss the limitations of this type of observational study (particularly the potential for confounding and selection bias inherent in the approach. The issues I have are only of clarity in a specific aspect - namely the identification of non-urgent attendances for the study. In the methods section it is not clear exactly how non-urgents are identified - the authors refer to those patients who are 'self referred' or 'referred by family or friends' and also to NEWS Score <1 and specific HRG codes. Did a non-urgent attendance have to fulfil all these criteria to be defined as such - this needs more clarity. There are other minor comments in the attached file. - The reviewer also provided a marked copy with additional comments. Please contact the publisher for full details.
---

REVIEWER	Geva Greenfield Imperial College London, UK
REVIEW RETURNED	23-Nov-2017

GENERAL COMMENTS	This is an important and well conducted study addressing a timely concern is the NHS and abroad and I support its publication. Some comments that can improve the manuscript are listed below. Strengths and limitations section. I suggest revising this section. It over exaggerates the impact of the study and sometimes incorrect. There are quite a few studies addressed this question before. The sentence "A& E are offering better continuity of care (perhaps more comprehensive care)" is speculative and unsupported by the findings. The background could benefit from a description of the way A/E and uccs work in the UK. It might be unclear to the international reader. It would be useful to specify what do you mean by "usual A/E staff", are they consultants, nurses etc.
---

	The heading “Main exposure variable: GP in A&E scheme” is confusing, perhaps differentiate between the intervention and the exposure viable used in your model. The discussion should be significantly elaborated. At the moment is pretty much replicates the findings section. It would be valuable to hear more about the ‘so what’ of your study and what are the challenges raised by the findings. Shouldn’t gps, for example, according to your findings, attend the A/E 24/7? Etc. Are there any downsides to having gps in A/E? In the limitations section it would be useful to mention that the study addressed only on wait times, but not adverse effects, mortality and patient outcomes other than 7d reattendance. References are there are only a few studied cited from the last 5 years. It would useful to add some more recent evidence. Minor comments: Abstract: It would be useful to emphasize that both the gps and emergency physicians worked in A/E. While it seems obvious, it is worth clarifying. It would be useful to add some numbers to the results section.
--	--

REVIEWER	Laura Anselmi Research Fellow, Manchester Centre for Health Economics, University of Manchester, UK
REVIEW RETURNED	27-Nov-2017

GENERAL COMMENTS	I enjoyed the paper and I think it addresses a very relevant and topical issue about what little evidence is still available. I find the conclusion reached sensible and somehow in line with what could be expected. I have three major concerns that I’ll discuss below and a few minor comments/requests for clarification. First, the study design may be the only feasible, but not the most appropriate to answer the research question. The comparison of patients triaged and the allocated to the care of a GP in A&E or standard care in A&E may be quite different in a way that is unobservable to the researcher. Unobservable differences in patient severity may bias the results of the analysis, particularly with respect to process outcomes. The authors are aware, and indeed discuss this as a limitation. However, I think it could be good to present results from sensitivity analyses based on alternative matching methods, for example matching with replacement, or matching based on alternative algorithms and/or patient characteristics. The authors could also justify why they choose a nearest neighbour algorithm with no replacement. Is it not possible to use data from before the introduction of GP in A&E? When were GP introduced in A&E in UHCW? The limitation linked to selection bias is acknowledged by authors. However, it could be discusses in more length. In particular the authors could outline what are the implications for results on process and clinical indicators and for their interpretation, distinguishing between the two groups. For example, is the reduced process outcomes for patients treated by a GP simply a reflection of lower severity? What do differences in clinical indicators mean? What do they say about patient safety overall? Most importantly, what are the next questions to be addressed? Given the nature of the two groups compared and the lack of the
--

	appropriate counterfactual (patients who should have been seen by a GP but as there is no one in A&E follow the standard A&E care route), I would question the use of the word evaluation in the title. I would suggest being more modest and simply saying that some evidence is presented. Other comments: Abstract: Indicate the time period covered by the study. Page 5: The first sentence looks incomplete. Methods: Would it be possible to provide a general description of the data set used, the variables included, perhaps after saying that UHCW data were used on page 5 line 38. Page 6, line 14: What is PMAF? It could be helpful to explain for the reader. Page 6 line 17: Would it be helpful to say that the patients arrive in three ways and clarify from which entrance self-referrals (the group used) come in? Page 6 line 50: what are the hours of operation of GP in A&E? Page 7 line 44: characteristics such as or exactly these characteristics? Page 9 line 12: How were minor cases identified? Based on HRG codes? Where changes are made these should be reflected in the STROBE checklist.
--	---

VERSION 1 – AUTHOR RESPONSE

Reviewer: 1 Reviewer Name: Colin O'Keeffe	
This was a well written, clearly described paper on a key area/topic for the NHS. The conclusions are valid and adequately discuss the limitations of this type of observational study (particularly the potential for confounding and selection bias inherent in the approach.	Thank you.
The issues I have are only of clarity in a specific aspect - namely the identification of non-urgent attendances for the study. In the methods section it is not clear exactly how non-urgents are identified - the authors refer to those patients who are 'self-referred' or 'referred by family or friends' and also to NEWS Score <1 and specific HRG codes. Did a non-urgent attendance	Thanks, this has been clarified. "These are unplanned visits to the A&E defined as those presenting with minor illness with a National Early Warning Score (NEWS) ≤ 1."

have to fulfil all these criteria to be defined as such - this needs more clarity.	
There are other minor comments in the attached file.	Thanks, we made all the suggested changes.
Reviewer: 2 Reviewer Name: Geva Greenfield	
This is an important and well conducted study addressing a timely concern is the NHS and abroad and I support its publication. Some comments that can improve the manuscript are listed below.	Thank your comments, which helped to improve the quality of the manuscript. We have addressed them as indicated below.
Strengths and limitations section. I suggest revising this section. It over exaggerates the impact of the study and sometimes incorrect. There are quite a few studies addressed this question before.	We have now toned down the sentence and now reference other studies in this area "Notwithstanding these limitations, the findings have some important policy and practice implications. Consistent with previous studies, these findings provide empirical evidence about GPs in A&E and A&E clinical indicators, supporting theories that .."
The sentence "A& E are offering better continuity of care (perhaps more comprehensive care)" is speculative and unsupported by the findings.	The sentence is supported by our finding that GP-in-ED were significantly more likely to refer more patients to out-patient "Our findings suggest that GPs are offering better continuity of care (perhaps more comprehensive care). We hypothesise that the fact that the GPs in A&E had access to primary care notes may be of importance here. Findings show that GPs made decisions more frequently about immediate onward referral to ambulatory clinics such as the urgent primary care clinic in UHCW or secondary care outpatient clinics. This is a more efficient use of resources avoiding both hospital admission and referral back to usual GP for them to make the referral. The removal of this unnecessary additional interaction reduces frustration for patients and doctors alike and should improve the patient experience. In addition, the prescribing data we examined could mean that GPs in A&E are fulfilling requests for repeat medications more often than A&E staff with the added safety of access to the primary care record again reducing the need for additional communications with surgery". Further work is necessary to examine the appropriateness and outcomes of these onward referrals and prescriptions.
The background could benefit from a description of the way A/E and uccs work in the UK. It might be unclear to the international reader.	We have now added the description to the background section "Accident and Emergency departments (A&E) departments in the UK offer access 24 hours a day, 365 days a year and the immediate team usually includes paramedics, A&E nurses, diagnostic radiographers, A&E reception staff, porters, healthcare assistants and emergency medicine doctors ¹ . Following arrival by ambulance or other self-determined mode of transport, the patient is triaged by a s qualified healthcare professional to ensure patients with the most serious conditions are seen first ¹ .

	Depending on the presenting symptoms and signs, patients can either be treated, transferred to a nearby urgent care centre, minor injuries unit or referred to a GP on site or discharged¹. Where treated, a patient is cared for by a team headed by a consultant in emergency medicine and the patients receive care from a suitably qualified member of the team.”
It would be useful to specify what do you mean by “usual A/E staff”, are they consultants, nurses etc.	We have now clarified this that they are medical staff who are specialists in all aspect of emergency medicine. “includes paramedics, A&E nurses, diagnostic radiographers, A&E reception staff, porters, healthcare assistants and emergency medicine doctors”
The heading “Main exposure variable: GP in A&E scheme” is confusing, perhaps differentiate between the intervention and the exposure viable used in your model.	Thanks, this has changed to “Intervention evaluated”
The discussion should be significantly elaborated. At the moment is pretty much replicates the findings section. It would be valuable to hear more about the ‘so what’ of your study and what are the challenges raised by the findings. Shouldn’t gps, for example, according to your findings, attend the A/E 24/7? Etc. Are there any downsides to having gps in A/E?	We have now elaborated the discussion section “As the bulk of A&E workload takes place Monday to Friday between 8-10pm there is a case for extending coverage of GP in ED between these hours but not 24 hours a day. Demand for primary care consultations at A&E locations reduces during the night and alternative primary care services are available. There are several workforce implications, however. Many organisations are seeking to train and recruit General Practitioners who can work in acute medicine settings in secondary and primary care with the expectation of developing a cadre of multiskilled healthcare professionals. There is perhaps also a case for extending opportunities for secondary care ED physicians to work in primary care during their training to develop some of the GP skillset which has less reliance on tests and a greater experience and knowledge of which cases can safely be managed outside of the hospital. However, recruitment and retention pressures in both specialities make this kind of collaborative training unlikely at present. There is also the opportunity for the primary care approach to diffuse into the ED – this is less likely in a self-contained, although co-located unit than locations where primary and secondary care clinicians work in a more integrated fashion There may be a dilution of primary care skills and approaches over time for a GP fully integrated in ED and their effectiveness will decrease over time as they become subsumed into the secondary care culture.. However, if secondary care seek to recruit less experienced GPs to fill these roles, this may be more of a problem, as they may not have sufficient primary care experience as an independent practitioner to maintain their professional skills and identity in a secondary care environment. A GP in A&E requires an accommodation of a culture which is used to handling uncertainty with a culture which seeks to reduce uncertainty in diagnosis and treatment of patients.”
In the limitations section it would be useful to mention that the study addressed only on wait times, but not adverse effects, mortality and patient outcomes other	We have already alluded to that in the study limitations “In addition, we do not know the eventual health outcome, although re-attendance data are reassuring on this front”

than 7d reattendance.	
References are there are only a few studied cited from the last 5 years. It would useful to add some more recent evidence.	We have now updated our reference to include three recent systematic reviews that summarised the existing evidence on interventions for reducing inappropriate use of A&E: 1. Ismail SA, Gibbons DC, Gnani S. Reducing inappropriate accident and emergency department attendances: a systematic review of primary care service interventions. The British journal of general practice : the journal of the Royal College of General Practitioners 2013;63(617):e813-20. doi: 10.3399/bjgp13X675395 [published Online First: 2013/12/20] 2. Ramlakhan S, Mason S, O'Keeffe C, et al. Primary care services located with EDs: a review of effectiveness. Emergency medicine journal : EMJ 2016;33(7):495-503. doi: 10.1136/emered-2015-204900 [published Online First: 2016/04/14] 3. Van den Heede K, Van de Voorde C. Interventions to reduce emergency department utilisation: A review of reviews. Health policy (Amsterdam, Netherlands) 2016;120(12):1337-49. doi: 10.1016/j.healthpol.2016.10.002 [published Online First: 2016/11/20]
Minor comments:	
Abstract: It would be useful to emphasize that both the gps and emergency physicians worked in A/E. While it seems obvious, it is worth clarifying.	This has now been clarified. "GPs in A&E co-located within ..." I also think we need to stress they are separate services
It would be useful to add some numbers to the results section.	We have now added numbers to the results section " Results: ... Compared with standard care in A&E standard care in A&E, GPs in A&E significantly: referred fewer patients to other specialists (risk ratio [RR] 0.10, 95% confidence interval [CI] 0.08 to 0.13), admitted fewer patients (RR 0.29, 95% CI 0.26 to 0.33), ordered fewer radiological investigations (RR 0.38, 95% CI 0.34 to 0.42), ordered fewer blood tests (0.54, 95% CI 0.50 to 0.60), and ordered fewer investigations (0.94, 95% CI 0.91 to 0.97). However, they intervened more, offered more primary care follow up (RR 1.78, 95% CI 1.68 to 1.89), and referred more patients to out-patient and other A&E clinics (RR 2.23, 95% CI 2.05 to 2.43). Patients seen by GPs in A&E were on average less likely to spend four hours plus in A&E (RR 0.35, 95% CI 0.28 to 0.42) compared with standard care in A&E . There was no difference in re-attendance after seven days (RR 0.93, 95% CI 0.82 to 1.07)."
Reviewer: 3 Reviewer Name: Laura Anselmi	
I enjoyed the paper and I think it addresses a very relevant and topical issue about what little evidence is still available. I find the conclusion reached sensible and somehow in line with what could be expected.	Thank you.
I have three major concerns that I'll discuss below and a few minor comments/requests for clarification. First, the study design	Thanks, as suggested, in this revised draft, we have now conducted series of sensitivity analysis "Methods

may be the only feasible, but not the most appropriate to answer the research question. The comparison of patients triaged and the allocated to the care of a GP in A&E or standard care in A&E may be quite different in a way that is unobservable to the researcher. Unobservable differences in patient severity may bias the results of the analysis, particularly with respect to process outcomes. The authors are aware, and indeed discuss this as a limitation. However, I think it could be good to present results from sensitivity analyses based on alternative matching methods, for example matching with replacement, or matching based on alternative algorithms and/or patient characteristics. The authors could also justify why they choose a nearest neighbour algorithm with no replacement. Is it not possible to use data from before the introduction of GP in A&E? When were GP introduced in A&E in UHCW?	To test the robustness of our primary findings we performed a series of sensitivity analyses. Firstly, we repeated the base case algorithm, nearest neighbour with replacement. Secondly, we also implemented another algorithm 'exact with and without replacement. Results The results of the sensitivity analyses are shown in eTable 4. When we used the replacement method, the magnitudes and directions of the effect estimates were similar to the primary analyses. Similarly, the results of an alternative matching algorithm 'exact' with and without replacement showed similar to the primary analyses.”
The limitation linked to selection bias is acknowledged by authors. However, it could be discusses in more length. In particular the authors could outline what are the implications for results on process and clinical indicators and for their interpretation, distinguishing between the two groups. For example, is the reduced process outcomes for patients treated by a GP simply a reflection of lower severity? What do	We have now discussed several implications for policy of study findings in the discussion section: “As the bulk of A&E workload takes place Monday to Friday between 8-10pm there is a case for extending coverage of GP in ED between these hours but not 24 hours a day. Demand for primary care consultations at A&E locations reduces during the night and alternative primary care services are available. NHS England’s chief executive and the chief executive of NHS Improvement recently recommended that “every hospital in England must have a ‘comprehensive’ GP led triage system in emergency department’ to provide a repeat of the winter crisis that usually grip the NHS service²². Two broad models of primary care services in A&E have been proposed ²³and these models are summarised in Box 1 ²³:  Box 1: Models of primary care services in A&E ²³  1. Re-direct – Patients present to the A&E and are sent to a primary care service:  a) Adjacent out-of-hours service. 

differences in clinical indicators mean? What do they say about patient safety overall? Most importantly, what are the next questions to be addressed?	 b) Adjacent walk-in-centre. c) Adjacent primary care/community service. d) Advice only/self-care. 2. Managing patients in the Emergency Department  a) Gatekeeping in A&E – Primary care service based at the front of A&E to manage patient entry to the A&E service. b) Primary care within A&E:  i. GP working in A&E:  1. Employed by primary care trust 2. Employed by Acute Trust. ii. Other primary care clinician. iii. A&E clinician. There are several workforce implications, however. Many organisations are seeking to train and recruit General Practitioners who can work in acute medicine settings in secondary and primary care with the expectation of developing a cadre of multiskilled healthcare professionals²⁴. There is perhaps also a case for extending opportunities for secondary care A&E physicians to work in primary care during their training to develop some of the GP skillset which has less reliance on tests and a greater experience and knowledge of which cases can safely be managed outside of the hospital²⁵. However, recruitment and retention pressures in both specialities make this kind of collaborative training unlikely at present. There is also the opportunity for the primary care approach to diffuse into the A&E – this is less likely in a self-contained, although co-located unit than locations where primary and secondary care clinicians work in a more integrated fashion There may be a dilution of primary care skills and approaches over time for a GP fully integrated in A&E and their effectiveness will decrease over time as they become subsumed into the secondary care culture. However, if secondary care seek to recruit less experienced GPs to fill these roles, this may be more of a problem, as they may not have sufficient primary care experience as an independent practitioner to maintain their professional skills and identity in a secondary care environment. A GP in A&E requires an accommodation of a culture which is used to handling uncertainty with a culture which seeks to reduce uncertainty in diagnosis and treatment of patients.”
Given the nature of the two groups compared and the lack of the appropriate counterfactual (patients who should have been seen by a GP but as there is no one in A&E follow the standard A&E care route), I would question the use of the word evaluation in the title. I would suggest being more modest and simply saying that some evidence is presented.	Thanks, the word evaluation has been removed from the title.
Other comments: Abstract: Indicate the time period covered by	This has been added “between May 2015 and March 2016.”

the study.	
Page 5: The first sentence looks incomplete.	This sentence has been corrected “Existing studies that have compared the effects of introducing GPs in A&E with standard care in A&E for managing non-urgent cases”
Methods: Would it be possible to provide a general description of the data set used, the variables included, perhaps after saying that UHCW data were used on page 5 line 38.	The variables were described under ‘outcomes and covariates’
Page 6, line 14: What is PMAF? It could be helpful to explain for the reader.	This has now been explained “PMAF was a £50m Challenge Fund to support practices to trial new and innovative ways of delivering GP services and making services more accessible to patients.”
Page 6 line 17: Would it be helpful to say that the patients arrive in three ways and clarify from which entrance self-referrals (the group used) come in?	Self-reported minor cases arrived via different methods, we did not stratify the analysis by mode of arrival. We have now simplified this sentence “The patients arrived at the A&E, were registered onto the Trust’s clinical A&E system (iPM) and ...”
Page 6 line 50: what are the hours of operation of GP in A&E?	Not specified, the service was operated at peak times.
Page 7 line 44: characteristics such as or exactly these characteristics?	They are exactly these characteristics and summary characteristics before and after matching are presented in Table 1.
Page 9 line 12: How were minor cases identified? Based on HRG codes?	This has been clarified, HRG codes were used to define minor cases (all).
Where changes are made these should be reflected in the STROBE checklist.	Thanks, we have now revised the STROBE checklist accordingly.

VERSION 2 – REVIEW

REVIEWER	Laura Anselmi Centre for Health Economics, Faculty of Biology, Medicine and Health, University of Manchester
REVIEW RETURNED	14-Jan-2018

GENERAL COMMENTS	The paper is clear and informative. After the revisions I still have some minor comments/ suggestions listed below. First, I would suggest to replace the term 'impact of' with 'evidence from' in the title. Second, on page 5 and perhaps somewhere else, the acronym ED is used as an alternative to A&E. It may be better to be consistent throughout the study. Third, after that the hours during which GPs operate have been specified, I believe you have two sources of selection bias due to unobservable patients characteristics (and severity). The first one is because patients that present at peak hours and at night may be different, while the second one is because the triage nurse decides who is to be treated in standard A&E or by a GP. It would be good to
--

	spell out clearly the two in the first paragraph of the limitations (unless it is specified in the study and I have missed it that only patients arriving at the same hours of the day are included in the sample for the analysis). Fourth, the discussion is now more substantial, but lack of references to prevent the reader thinking that they report the authors opinion on the best way to organize services.
--	--

REVIEWER	Geva Greenfield Imperial Collge London
REVIEW RETURNED	02-Feb-2018

GENERAL COMMENTS	This is much improved version – thank you. Two last comments: I am still not convinced by the sentence “A& E are offering better continuity of care (perhaps more comprehensive care)” in what sense did anything you have fund related to continuity of care? It is yet remains unclear. It would be useful to define what you mean by continuity of care and how your findings support to it. “Findings show that GPs made decisions more frequently about immediate onward referral to ambulatory clinics such as” – I’d probably choose another word than ‘frequently’ because it is rates, not the frequency that are measured.
---

VERSION 2 – AUTHOR RESPONSE

Reviewer: 2 Reviewer Name: Geva Greenfield	
This is much improved version – thank you. Two last comments:	Thanks for your comments.
I am still not convinced by the sentence “A& E are offering better continuity of care (perhaps more comprehensive care)” in what sense did anything you have fund related to continuity of care? It is yet remains unclear. It would be useful to define what you mean by continuity of care and how your findings support to it.	Thanks, we have changed it to “more comprehensive care”
“Findings show that GPs made decisions more frequently about immediate onward referral to ambulatory clinics such as” – I’d probably choose another word than ‘frequently’ because it is rates, not the frequency that are measured.	We have changed this to “Findings show that GPs are more likely to make immediate onward referral”
Reviewer: 3 Reviewer Name: Laura Anselmi	
The paper is clear and informative. After the revisions I still have some minor comments/ suggestions listed below.	Thanks for your comments.
First, I would suggest to replace the term 'impact of' with 'evidence from' n the title.	There is no 'impact' in the title “General practitioners providing non-urgent care in emergency department: a natural experiment”
Second, on page 5 and perhaps somewhere else, the acronym ED is used as an alternative to A&E. It may be better to be consistent throughout the study.	Thanks, we have changed the only occurrence of ED to A&E.

Third, after that the hours during which GPs operate have been specified, I believe you have two sources of selection bias due to unobservable patients characteristics (and severity). The first one is because patients that present at peak hours and at night may be different, while the second one is because the triage nurse decides who is to be treated in standard A&E or by a GP. It would be good to spell out clearly the two in the first paragraph of the limitations (unless it is specified in the study and I have missed it that only patients arriving at the same hours of the day are included in the sample for the analysis).	We have now included 'ArrivalHour' as one of the matching variables; and the magnitudes and directions of the effect estimates were very similar to all the previous analyses.
Fourth, the discussion is now more substantial, but lack of references to prevent the reader thinking that they report the authors opinion on the best way to organize services.	We cited two systematic reviews that summarized evidence from hundreds of studies on best models to organize GPs working in A&E. In addition to summarizing the evidence from these reviews, we also added a box summarizing the two broad categories of models of primary care services in A&E.